# Therapeutic Approaches in Lysosomal Storage Diseases

**DOI:** 10.3390/biom11121775

**Published:** 2021-11-26

**Authors:** Carlos Fernández-Pereira, Beatriz San Millán-Tejado, María Gallardo-Gómez, Tania Pérez-Márquez, Marta Alves-Villar, Cristina Melcón-Crespo, Julián Fernández-Martín, Saida Ortolano

**Affiliations:** 1Rare Disease and Pediatric Medicine Group, Galicia Sur Health Research Institute (IIS Galicia Sur), SERGAS-UVIGO, 36312 Vigo, Spain; carlos.fernandez@iisgaliciasur.es (C.F.-P.); beatriz.san.millan.tejado@sergas.es (B.S.M.-T.); maria.gallardo@iisgaliciasur.es (M.G.-G.); tania.perez@iisgaliciasur.es (T.P.-M.); marta.alves@iisgaliciasur.es (M.A.-V.); cristina.melcon.crespo@sergas.es (C.M.-C.); jorge.julian.fernandez.martin@sergas.es (J.F.-M.); 2Department of Pediatrics, Hospital Álvaro Cunqueiro, SERGAS, 36213 Vigo, Spain; 3Department of Internal Medicine, Hospital Álvaro Cunqueiro, SERGAS, 36213 Vigo, Spain

**Keywords:** lysosomal storage diseases, enzyme replacement therapy, gene therapy, small molecules, autophagy

## Abstract

Lysosomal Storage Diseases are multisystemic disorders determined by genetic variants, which affect the proteins involved in lysosomal function and cellular metabolism. Different therapeutic approaches, which are based on the physiologic mechanisms that regulate lysosomal function, have been proposed for these diseases. Currently, enzyme replacement therapy, gene therapy, or small molecules have been approved or are under clinical development to treat lysosomal storage disorders. The present article reviews the main therapeutic strategies that have been proposed so far, highlighting possible limitations and future perspectives.

## 1. Introduction

### 1.1. General Considerations concerning Rare Diseases

The last 25 years have been characterized by an upgrowing interest among the scientific community towards rare diseases and the development of specific treatments for these pathologies. This initiative finds its impulse in the increased sense of social and ethical responsibility to provide therapeutic solutions to the affected groups, as well as in the rising efforts of the pharma industry to dedicate resources to this field of knowledge.

The discovery of orphan drugs is a matter of interest for the industry, since most rare diseases exhibit well defined pathophysiological mechanisms, as they are congenital pathologies that affect a single gene. In addition, the process of approving orphan drugs comes with a specific designation and a relatively less complex process to facilitate possible solutions for unmet medical needs.

Additionally, an orphan drug can potentially have secondary indications directed to the treatment of frequent pathologies, with more complex etiology, which share molecular mechanisms with the rare disease for which the drug was originally developed.

For all these reasons, in the last years, the development of treatments orientated to rare diseases has largely increased, also favoring innovation of the biological tools and advances in the knowledge of disease natural history.

These considerations are particularly applicable to the field of Lysosomal Storage Diseases (LSDs) that show favorable characteristics for different therapeutic approaches.

### 1.2. Molecular Basis of LSDs and Possible Therapeutical Strategies

LSDs are caused by mutations in genes which encode for acid hydrolases, integral membrane proteins, activators and transporter proteins, or other proteins involved in lysosomal function. Deficiencies of these molecules unleash a metabolic imbalance, which results in substrate accumulation in multiple organs or tissues [1].

The therapeutic options currently available or under development for LSDs are based on the physiology of the lysosome, which is considered as a master regulator of cell metabolism.

Lysosomes are organelles principally dedicated to cell catabolic function through the action of more than 60 hydrolytic enzymes contained inside their lumen. However, nowadays we know that the lysosome is also important in the regulation of anabolic processes. In fact, it acts as a nutrient sensor and it regulates signal transduction, cellular growth, and vesicle trafficking [2].

These processes are widely related to the activity of the mammalian target of rapamycin complex 1 (mTORC1) of proteins, the principal regulator of cell growth, which is anchored to lysosomal membrane and senses variations in cytoplasmic amino acid concentration. At low amino acid concentrations, mTORC1 triggers autophagy, a process by which the cells sequester their own content to degrade it and recycle its components [3,4].

Moreover, the expression of genes that encode for lysosomal proteins depends on the action of the transcriptional factor EB (TFEB), which controls lysosomal biogenesis and exocytosis, lipid catabolism and energy metabolism, allowing the adaptation of the lysosome to different pathophysiological conditions [5]. The mTORC1 regulates TFEB by phosphorylation and retains it in the cytoplasm. Dephosphorylation of TFEB occurs in fasting conditions, following mTORC1 inactivation (Figure 1). Therefore, this mechanism is a proof of lysosome to nucleus communication and supports the concept of the lysosome as the main actor in the regulation of cellular metabolism [6].

Lysosomal enzymes are produced in the cytoplasm and migrate to the lysosome through well-established endocytic routes. In most cases, enzyme functionalization occurs in the Golgi apparatus by the addition of mannose-6-phosphate (MP-6) groups, which enables their entrance in the lysosome through the MP-6 receptor (M6PR) [7]. Nonetheless, some enzymes use different mechanisms; for instance, β-glucocerebrosidase, which reaches the lysosome through the lysosomal integral membrane protein-2 (LIMP-2) receptor [8].

Vesicles transport of substrates to the lysosome occurs throughout a variety of endocytic mechanisms (phagocytosis, macropinocytosis, and endocytosis, independent or mediated by clathrin or caveolin) determined by vesicle content [9]. By this means, substrate accumulation generated in LSDs determines the blockage of the autophagic process, as described in different disorders.

In Pompe Disease (PD), the autophagic compartment has been reported to be expanded in muscular tissues [10], and in Multiple Sulfatase Deficiency (MSD), all sulfatases are affected by a post-translational defective activation of the sulfatase modifier factor 1 that impairs autophagolysosome formation [11]. Finally, in Niemann Pick disease C (NPC), sphingosine storage leads to calcium increased concentration that blocks autophagy and increases the accumulation of cholesterol, sphingomyelin, and glycosphingolipids [12].

Therefore, the recently discovered connection between many LSDs, such as Gaucher disease (GD) and NPC, and other conditions, like ageing or neurodegenerative disorders (e.g., Parkinson and Alzheimer Disease), is easier to explain considering that these diseases are also caused by imbalance in the lysosome-autophagy axis [13].

On the other hand, substrate accumulation may also trigger an anomalous activation of other routes, such as the activation of a Toll-like receptor 4 (TLR-4) by Globotrialosil ceramide (Gb3) in Fabry disease (FD), which determines the response of the innate immune system [14] and endothelial disfunction [15].

Physiology of the lysosome is at the basis of all therapeutic strategies developed so far for LSDs, which appoints to different targets or approaches (Figure 2), detailed as it follows:Enzyme replacement therapy (ERT) consists of the intravenous administration of a properly glycosylated and functional form of the enzyme impaired in the disease, and takes advantage of the MP6R and the endogenous endocytic routes to reach the lysosomes within the target cells.Ex vivo gene therapy aims to administer a functional enzyme through the autologous transplant of hematopoietic stem cells, which are genetically modified in vitro.In vivo gene therapy consists of the direct injection of non-replicating non-hazardous viral vectors, which encode for the functional enzymes of interest in the transduced cells.Pharmacological chaperones (PC) stabilize the mutated enzymes to avoid their degradation in the endoplasmic reticulum and therefore, facilitate their translocation to the lysosome and the degradation of accumulated substrates.Substrate reduction therapy (SRT) and autophagy regulating drugs aim to target the molecular pathways in which the mutated protein participates, trying to restore the correct balance between synthesis and degradation of the substrates by slowing down the synthesis, accelerating the degradation or regulating vesicle trafficking of the substrates.


## 2. Enzyme Replacement Therapy

### 2.1. Available Drugs and ERT Mechanism of Action

ERT is currently the gold standard for the treatment of LSDs, since it has been clinically available for approximately two decades [16]. The development of these drugs was possible once the mechanisms of synthesis and transport of lysosomal enzymes were elucidated, and the efficacy of ERT was proved for the first time in 1995 in GD patients [17].

Nowadays there are different commercialized enzymes for GD, FD, PD, mucopolysaccharidosis I (MPS) I, MPS II, MPS IVA, MPS VI, and MPS VII, while many others are under development (Table 1).

As previously mentioned, ERT is based on the intravenous administration of a functional recombinant human enzyme, which can compensate the consequences of the congenital defect. ERT exploits the mechanism of lysosomal enzyme biogenesis and recycling. The recombinant protein, enriched with M6P groups, enters the cell using the MP6R receptor and reaches the lysosome. A wider distribution of the enzyme is possible thanks to the phenomenon of the cross-correction, which was described for the first time by Barton and Neufeld [26]. They observed that cells derived from LSD patients were able to metabolize the accumulated substrates, when co-cultivated with cells from healthy donors, which is possible because a fraction of lysosomal enzymes is secreted towards the extracellular space and enters the surrounding cells through the MP6R.

Recombinant human enzymes are produced in stable cell lines as Chinese Ovary Hamster (CHO) cells or human fibroblasts, which provide different glycosylation patterns [27]. Recently, vegetable cells (e.g., tobacco derived cells) [28] or yeasts [29] have also been used to produce the recombinant enzyme with lower production costs.

ERT has shown its efficacy in delaying the progression of the disease and improving the life quality of the patients. As an example, type 1 GD patients respond fairly well to ERT that can easily reach the macrophage to cleave glucosylceramide storage. Macrophages are responsible for the major clinical signs in this condition; nonetheless, ERT is not effective in neurological manifestations of type III GD and differentially distributed to tissues that are more difficult to target, such as the bone [18]. In FD, ERT reduces Gb3 levels in plasma and tissues, improves gastrointestinal symptoms and neuropathic pain and delays disease progression, by partially stabilizing heart and renal function [30].

ERT effectively reduces urinary glycosaminoglycans (GAGs), and liver and spleen volume in MPS patients (MPS I, II, VI and IVA), while cartilaginous organs such as trachea, bronchi, bones, eyes, and cognitive impairment, are poorly impacted by ERT [31].

### 2.2. Limitations of Enzymatic Replacement Therapy

Despite its undoubtable benefits, ERT entails different limitations that new generation enzymes aspire to improve.

One of the major drawbacks of ERT is the low half-life of the recombinant injected enzyme, which obligates to frequent administrations (usually biweekly infusion) during the whole life of the patient. Hence, enzyme concentrations in plasma fluctuate due to the fast degradation of recombinant proteins, which causes an On-Off effect in the therapeutic pattern.

In addition, the drug bioavailability is variable, since organs such as the liver and spleen, which express high MP6 receptor levels, tend to sequester the majority of the administered dose [32], while other organs are more hardly targeted. The enzymes have limited expression in bones, cartilage, or eyes, and are excluded from the central nervous system (CNS), as they are not able to cross the blood brain barrier (BBB).

To overcome this problem, intrathecal and intraventricular routes of administration are being tested to treat LSDs with neurological involvement (e.g. MPS I and VI, metachromatic leukodystrophy, MLD, etc.) by ERT and special equipment for the continuing administration of the enzymes to the CNS have been also developed [33,34]. Nonetheless, this is a manageable solution for patients with LSDs and CNS impairment.

Finally, is it possible that the patients treated with ERT can develop IgG antibodies against the recombinant enzyme. These antibodies can either be of neutralizing type, which directly binds to the enzyme and suppresses its catalytical function, or of non-neutralizing type, promoting enzyme elimination from immune cells throughout the Fc receptors [35]. Both mechanisms have been observed in patients under ERT treatment [36].

Complete neutralization of ERT effects was observed in PD patients, who do not express acid α-glycosidase. These patients are known as cross-reactive immunologic material (CRIM) subjects and may present a strong immune response to ERT [37], which can be attenuated by immunomodulation [38]. Allergic reactions with IgE antibody production has also been reported in a few cases [39].

### 2.3. New Generation Enzyme Replacement Therapy

New generation ERT attempts to produce more stable recombinant enzymes, using liposomes or nanoparticles envelops, which can enhance the half-life of the recombinant protein in the blood. Nanoparticles such as polystyrene or polyelectrolyte capsules, liposomes or extracellular vesicles are being tested [40].

Among the new generation enzymes, pegunigalsidase (PRX-102) is in advanced stage of development for the treatment of FD. PRX-102 is a recombinant α-galactosidase A covered by covalently bounded polyethylene glycol (PEG) moieties produced in tobacco derived cells. The PEG coating slows down protein degradation and facilitates the binding between enzyme monomers [41]. PRX-102 has been recently tested in an open label Phase I/II clinical trial (NCT01678898), involving 18 patients with FD. The drug was administered by intravenous infusion every 2 weeks during 12 months at three doses (0.2 mg/kg, 1 mg/kg, and 2 mg/kg) and enzyme levels were constantly maintained in the blood, while Gb3 and Lyso-Gb3 decreased [42]. In the extension study (NCT01769001), the drug was well tolerated with only moderate side effects. Following 24 months, gastrointestinal symptoms and Lyso Gb3 levels significantly decreased. Renal and cardiac functions remained stable after treatment, based on eGFR and LVMI data, respectively. Currently, PRX-102 efficacy is being evaluated in three phase III clinical trials in patients who were previously treated with agalsidase alfa (NTC03018730) or algasidase beta (NTC02795676 or NCT03180840).

The recombinant proteins used in ERT can also be functionalized with groups that allow receptor mediated internalization of the enzyme in the target cells, independently of its glycosylation pattern. These approaches to reach difficult targets, such as the CNS, recalls the Trojan Horse strategy and, therefore, the functional groups that are used are often denominated as *trojans*. These functional groups (i.e., recognized signal peptides by the insulin or transferrin receptor) allow to the recombinant enzymes to trespass the BBB. Examples of this strategy with good preclinic results have been described for MPS IIIA, a neurodegenerative disease that affects the Sulfoglucosamine Sulfamidase (SGSH) enzyme. It has been demonstrated that the chimeric forms of SGSH bound to antibody fragments against, respectively, transferrin or insulin receptor, are expressed at therapeutic levels in the brain of *knock out* mice or primates (rhesus monkey) and it can significantly reduce GAG accumulation [43].

## 3. Hematopoietic Stem Cell Transplantation

Hematopoietic stem cell transplantation (HSCT) was the main therapeutic option for LSDs before ERT became available, and it is still a useful strategy in certain disease with CNS involvement.

Indeed, an important advantage of HSCT is that donor-derived cells that produce functional enzymes are able to migrate to the brain, thus delaying neurocognitive degeneration [44]. Improvement in CNS manifestations and life quality is more efficiently achieved when the transplant is performed at an early stage of the disease. HSCT is indicated for the treatment of MPS I, in Krabbe disease, and in the attenuated forms of metachromatic leukodystrophy. In severe MPSI, HSCT increases life expectancy and improves clinical symptoms in children; therefore, it is the preferred treatment for patients diagnosed before the age of 2.5 years [45]. In Krabbe disease and MLD, the disease phenotype and stage of disease progression are of fundamental importance in determining successful outcomes [46,47].

However, the most important limitations of HSCT are the high morbidity and mortality rates of the process, related to rejection and infections, which, together with the variables levels of cell engraftment, eventually limits the applicability of this strategy to a few cases.

## 4. Gene Therapy in LSDs

Rare diseases, and LSDs in particular, have been an object of study for the development of gene therapy strategy due to their favorable characteristics. LSDs are mostly caused by mutations in genes encoding hydrolytic enzymes, which, due to their inherently catalytic nature, do not need to be expressed at high levels to determine therapeutic effects. It has been described that storage accumulation does not occur if residual activity is about 10% of physiologic values [48]. In addition, the cross-correction phenomenon helps to avoid substrate accumulation in different target organs, although not all cells are transduced by gene therapy vectors.

Currently, the only approved gene therapy for LSDs is OTL-200 for the treatment of MLD, however there are several candidates that are being tested in clinical trials and which will possibly be available at short-term.

### 4.1. Ex Vivo Genetic Therapy

Ex vivo gene therapy is based on a similar approach to allogenic transplantation of stem cells of hematopoietic origin (CD34^+^); however, in this case, the re-implanted cells are extracted from the affected patient and the genetic defect is corrected in vitro. Genome editing in extracted cells is possible through the action of endonucleases (TALEN system, Zinc-finger nucleases, and the CRISPR-Cas9) that specifically cut the genomic DNA in the mutation locus [49,50]. These systems allow a directed homologous recombination and, therefore, reduce the risk of mistarget in unrelated loci.

Viral vectors (e.g., AAV, Lentivirus, Adenovirus, etc.) are usually used to drive the editing system into target cells, however, liposomes or other type of nanoparticles can also vehicle these DNA sequences.

Compared to the allogenic transplant of hematopoietic cells, ex vivo gene therapy has the advantage of allowing autologous cell reimplantation and, therefore, can avoid immune rejection. Moreover, it maintains the favorable regenerative power of stem cells in damaged tissues.

Given its recent development, gene editing tools are continuously improving to overcome the limitations that still concern the application of these techniques. In fact, there is still a necessity to improve the percentage of corrected cells that are efficiently delivered. Moreover, reimplantation protocols need to be preceded by aggressive administration of immunosuppressive drugs to facilitate engraftment.

In addition, mistargeting is still a possibility as the guide RNAs may include short sequences that could be repeated along the whole genome. This risk obligates the need to sequence the genome of in vitro treated cells and select for the correctly modified clones.

Among the examples of successful ex vivo gene therapy drugs, approved or in the clinical trial phase (Table 2), there are Lentiviral vectors for the treatment of MLD [51] or FD [52].

In MLD, the phase III clinical trial (NCT04283227) assessed the pharmacodynamic effects and the long-term safety and efficacy of OTL-200 in late juvenile patients. The main end points were focused on measuring human arylsulfatase A (ARSA) activity in cerebrospinal fluid (CSF) and neuronal metabolite *N*-acetyl-aspartate (NAA) to creatinine (Cr) ratio, in white matter regions of the brain [53,54].

AVR-RD-01 is a Lentivirus based ex vivo gene therapy for FD. The first data obtained from the phase I/II clinical trial (NCT02800070) showed no serious adverse effects in the five enrolled male patients. All subjects presented increased α-GalA activity levels following intravenous infusion. After reaching a peak, activity levels decreased but did not return to null values (1 nmol/h/mL) [55]. In the extended study (NCT03454893), AVR-RD-01 showed similar results in producing functional enzyme and reducing Gb3 levels, as well as achieving a controlled eGFR state.

A similar strategy was developed for ex vivo gene therapy of GD and is now under evaluation (NCT04145037, phase I/II). The trial compares the results from naïve patients versus treated patients, evaluating outcomes like glucocerebrosidase activity levels and concentration of Lyso-GL1 [56].

In MPSI, phase I/II clinical trial (NCT03488394), aims to transduce the human α-L-iduronidase gene in cells from eight participants (28 days–11 years). The main endpoints of the trial are the effective hematological engraftment achievement, the overall survival and the normalization of urinary GAGs levels [57].

On the other hand, in MPSII, the phase I/II clinical trial (NCT03566043) aims to assess a dose regimen for efficient delivery of functional iduronate-2-sulfatase gene to the CNS of 12 enrolled patients [58].

### 4.2. In Vivo Genetic Therapy

In vivo gene therapy is based on direct injection of a viral vector encoding the gene of interest. Therefore, vectors that integrate into the genome are not usually employed for this technique, due to the difficult control of the transgene insertion locus in the genome.

Adeno-Associated Virus (AAV) are the most used vectors for in vivo gene therapy. AAVs are viruses from the *Dependoparvovirus* genus of the *Parvoviridae* family that need co-infection with an adenovirus or herpes virus for their replication [59]. Moreover, in AAV-based vectors, the viral genome is almost entirely replaced by the transgene and the only viral regions that are conserved are the ones that allow viral particles to enter the cells (Inverted Terminal Repeats), while capsid and replication related genes are removed. Viral particles assembly is only possible in vitro through the co-transfection of the vector of interest with other plasmids that express the Rep and Cap genes, as well as the *Helper* gene from the adenovirus [60].

In vivo gene therapy injected vector remains in the cell in the form of episome (circular DNA fragment) and takes advantage of the cell machinery to produce the protein encoded by the transgene.

Therefore, this therapy is similar to ERT, but the continuous production of recombinant protein ensures higher half-life and bioavailability of the enzyme of interest. In fact, with a single injection of the viral vector, sustained expression of the recombinant protein can be observed in many tissues.

Different serotypes of AAV with specific tissue tropisms have been described, including serotypes that are able to transduce the BBB [61].

Although it is still unknown if the injection of AAV vectors can cover life-long expression of the transgene, preclinical studies have demonstrated that these vectors drive long-term expression of the transgene in animal models [62].

The AAV can also be employed to deliver gene editing tools, which drive safe integration of the transgenes in human genome. This is the principle applied in the liver directed therapy that is being developed for different LSDs (i.e., PD and FD) [63,64,65,66]. In this strategy, AAVs transport gene editing tools and the transgene to a safe locus (i.e., albumin locus) in the liver, where the proteins of interest are selectively produced through the action of hepato-specific promoters. The recombinant enzyme is redistributed to other organs, thanks to the cross-correction effect (Figure 3).

AAV vectors are currently being tested in clinical trials for the treatment of LSDs (Table 3).

In PD, a phase I/II clinical trial (NCT04093349) seeks to evaluate a SPK-3006 single dose in 20 participants in order to test safety and possible immune response against AAV capsid, as well as effective acid-α-glucosidase production [67].

FLT190 is an AAV8 based vector driving α-Galactosidase A expression to the liver, which is currently being tested (NCT04040049, phase I/II) for the treatment of FD. The trial aims to evaluate safety and efficacy in 12 enrolled patients [68,69].

A similar approach is used by ST-920, which is an AAV2/6 based vector driving *GLA* expression in the liver. This vector showed efficacy in a FD mouse model [66] and is currently being evaluated in a phase I/II clinical trial (NCT04046224).

In MPS I, a first-in-human phase I/II clinical trial (NCT03580083) is intended to deliver a functional copy of the α-L-iduronidase in the CNS, through an in vivo injected AAV9 vector (RGX-111). RGX-111 is intramuscularly injected to assess the dose in five enrolled patients. The main outcomes focus on neurodevelopmental parameters [70].

A similar vector has been developed for the treatment of MPSII and it is being tested in a phase I/II clinical trial (NCT04571970, NCT03566043, NCT04597385) to assess safety and long-term follow-up. Outcomes include GAG levels and iduronate-2-sulfatase activity [70].

AAV vector SAF-301 has been developed for the treatment of MPS III A and its long-term safety and efficacy are under evaluation in a phase I/II clinical trial (NCT02053064) after intracerebral injection of the vector in 4 participants [70].

Finally, vectors based respectively on AAV9, AVVrh10, and Hu68 serotypes, have been developed to treat Krabbe Disease. Positive preclinical outcomes were reported in preclinical studies on dog models, upon direct injection of the drug in the CNS [71,72]. Two of these gene therapy tools are going to be tested in clinical trials that started to recruit patients in September 2021 and aim to deliver functional galactosylceramidase in the CNS (NCT04771416, NCT04693598).

Nonetheless, AAV-based in vivo gene therapy still presents some limitations to overcome. One of the major drawbacks of AAV is the viral particle size, which only allows the inclusion of DNA fragments of the maximum size of 5Kb, restricting this application to small genes. However, by combining different constructs and editing tools, AAV vectors can also be eventually suitable to correct genetic defects in larger genes.

Perhaps the major limitation when using in vivo gene therapy is the pre-existing immunity towards AAV [45]. These viruses were isolated for the first time from human tissues, and the majority of the population produce antibodies against their capsids, which may limit the efficacy of this strategy, especially when it comes to AAV1 and AAV2, which are the most frequently detected serotypes. Therefore, in an attempt to by-pass pre-existing immunity, the most used serotypes, up to date, are the least abundant in nature (e.g., AAV8, AAV9, and AVV10) or the recently developed AAVs with chimeric capsids. However, antibody production against the virus can also be controlled by employing immunosuppressive drugs, before vector administration.

Synthetic capsids also provide a promising solution to control the immune response in case re-injection is required when episomal DNA is lost, as a consequence of repeated cell cycles.

## 5. Small Molecules of Oral Administration

The so-called small molecules are synthetic compounds of low molecular weight that can be orally administered and do not entail immune system activation. Small molecules can diffuse through cell membranes and reach different tissues, including the CNS.

Up to date there are several small molecules in use, or under clinical development, for LSD treatment (Table 4).

Small molecules can be used to treat LSDs throughout different mechanisms of action:pharmacological chaperonessubstrate reduction therapiesautophagy and proteostasis regulators

Moreover, certain molecules are potentially effective in more than one LSDs, since these pathologies share metabolic routes [77] (Figure 4).

### 5.1. Pharmacological Chaperones

Pharmacological chaperones are small molecules, which proved to be useful in patients with mutations affecting the protein folding. These molecules bind to the affected enzyme, stabilizing its structure during its synthesis. As a consequence, the mutated enzyme avoids the degradation mechanisms at an endoplasmic reticulum level and can be transferred to the lysosome. In the lysosome, the chaperone is released due to the low pH and the high substrate concentration, so the enzyme can catalyze substrate cleavage, even though the efficiency of the reaction is partially reduced [78].

Therapy with PCs is available for FD and is under evaluation for other LSDs. Dr. Fan et al. synthetized 1-deoxygalactonojirimycin (DGJ or Migalastat) a competitive inhibitor of α-Galactosidase A, which is the first approved PC for FD [79].

The efficacy of this drug is comparable to that of ERT, and allows meliorating parameters related to cardiac and renal functions [73,80].

However, the major limitation of Migalastat is that it is effective only with specific mutations of the enzyme. Indeed, these kinds of molecules cannot be effective in absence of the target protein caused by large deletions, splicing variants or frameshift mutations. The response to PCs is also variable in single amino acid substitutions, depending on the position of the residue. It is reasonable to predict that patients with mutations affecting catalytic activity, or located far from the binding site of the molecule, are not willing to benefit from the treatment. For this reason, Migalastat is approved only for patients who present amenable mutations (about 40–60% of diagnosed FD). In order to foresee the response of FD patients to Migalastat, both in silico and in vitro tests have been developed and clinically validated to predict the susceptibility of specific mutations to the drug [81].

Specifically designed PCs may also be useful for GD treatment, which is caused by p.Asn370Ser mutation in about 70% of the cases or by p.Leu444Pro variant in 38% of the affected persons, whereas in PD it was estimated that 10–15% of known mutations can be susceptible to PCs [82,83].

Nevertheless, not all enzymes involved in LSDs respond sufficiently well to the action of PCs. In fact, 1-deoxynojirimycin (DNJ) was successfully tested in mice models, where it proved to enhance α-Glycosidase stability (enzyme causing PD), but these effects were considerably lower in primates and in leukocytes from healthy human subjects.

One way of enhancing the efficacy of the PCs is by applying a discontinuous administration regime. This is because the half-life of the enzyme is a matter of days whereas the PC is a matter of hours. When the drug is in the bloodstream, the traffic of the enzyme to the lysosome is maximized, while, after its concentration decays, substrate metabolization is enhanced, favoring chaperone release at the lysosome. In FD mice models the discontinuous administration (alternate days) of Migalastat resulted in a major storage metabolization compared with the daily drug administration [83].

Furthermore, it is possible that new generation PCs, which interact with allosteric sites of the protein, may better stabilize the protein and increase the spectrum of susceptible mutations. These molecules would not need to be released by the protein and would further maintain the correctly folded structure of the protein. Possible therapeutic targets with allosteric function have been identified in α-Galactosidase, in acid α-Glycosidase and β-Glycosidase [84,85,86].

Currently, the co-administration of PCs with ERT is under investigation in order to evaluate the possible synergic effect of both treatments. In fact, PCs can also bind to recombinant enzymes, stabilizing them and favoring their transit towards the lysosome. Data that confirm this hypothesis have been obtained in vitro and in vivo for Migalastat [87] and for other experimental PCs for LSDs treatment [88].

### 5.2. Substrate Reduction Therapy

SRT consists of the use of small molecules to inhibit the enzyme that synthetizes the substrate (or one of its precursors) accumulated in a specific LSDs, to restore synthesis/degradation balance.

Miglustat and Eliglustat are competitive inhibitors of the glucosylceramide synthetase developed as SRT for GD treat GD [77].

Miglustat is an oral drug, conditionally approved for GD patients who cannot tolerate ERT, because it seems to act less efficiently than the recombinant enzymes. Despite of its capacity to cross BBB, Miglustat can only be used for GD Type I patients, due to its potentially neurotoxic effects, as it is an agonist of the glucose sensor SGLT3, which is expressed at the neuromuscular junction [89]. Moreover, Miglustat may also cause less severe side effects like diarrhea, abdominal pain, and weight loss, due to the impaired intestinal absorption of disaccharide determined by the inhibition of glycosidase isomers at an enteric level [90]. In spite of these limitations, Miglustat proved to be effective to improve cognitive and motor function in NPC patients, since it decreases glucosylceramide synthesis and it is used for the treatment of this disease [76].

On the other hand, Eliglustat proved to be effective and comparable to ERT in non-treated GD I patients [75]. Eliglustat does not inhibit intestinal glycosidases neither is it neurotoxic. Nonetheless, this drug is a substrate of the Glycoprotein-P transporter and can compete with other substrates of this receptor (e.g., drugs such as digoxin, phenytoin, and colchicine) which can affect its pharmacokinetics [91]. In addition, this active principle is metabolized by the P450 cytochrome and can interact with other drugs eliminated through this route, so pharmacokinetics of Eliglustat can also be affected by the genetic heterogenicity of CYP2D6 enzyme at the P450 cytochrome. For this reason, the dose of this treatment has to be adjusted according to the kind of CYP2D6 enzyme expressed by the subject, which determines a patient’s status of extensive, intermediate, or poor metabolizer.

SRT is currently under development also for the treatment of FD. Lucearstat is an iminosugar which inhibits the production of glucosylceramide synthase and, therefore, the synthesis of Gb3. In phase I clinical trials (NCT02930655), Lucerastat was able to reduce Glucosphyngolipids accumulation (GL-1,Gb3 and lactosilceramide) [92]. Currently, the molecule is under evaluation in phase III clinical trial MODIFY (NCT03425539), an aleatory double-blind placebo controlled study to assess safety and efficacy [92,93].

Venglustat is a new SRT drug, which inhibits glucosylceramide synthase (GCS) and so it reduces glucosylceramide synthesis [94]. This drug has been tested for different therapeutic indications and appears to be useful for the treatment of FD. A phase II clinical trial (NCT02489344) assessed long-term safety of Venglustat (GZ/SAR402671) in eight adult male patients with FD, finding reduced levels of Gb3, Lyso-Gb3 and monosialodihexosylganglioside (GM3) in treated patients.

### 5.3. Autophagy and Proteostasis Regulatory Molecules

Proteostasis is a complex network that regulates the synthesis, folding, trafficking, aggregation, and the degradation of proteins in the cell. Proteostasis regulators (Celastrol and MG-132), especially if associated with a PCs, can be an alternative for LSDs treatment as they facilitate the enzyme transit towards the lysosome. These compounds were assayed in vitro for GM2 Gangliosidosis (Tay-Sachs disease) and GD treatment [95].

In addition, targeting autophagy related pathways may help to contrast storage accumulation in LSDs, by reactivation of substrate degradation. The advantage of this strategy is that the same treatment can ideally be effective for different LSDs, independently of the genetic defect.

As an example, it has been shown that the stimulation of Ca^+2^-dependent exocytosis mediated by TFEB increases the metabolization of accumulated substrates in MSD and in PD [96]. Medina et al., described how TFEB overexpression in a PD mouse model reduces the glycogen storage and lysosome size, improving the autophagolysosome processing and reducing the excessive presence of autophagic vesicles. A similar strategy was proved in fibroblasts of patients with GD [97].

Genistein is a natural isoflavonoid that has been successfully tested to stimulate the autophagy through TFEB activation. As a consequence, GAGs levels were find to be reduced and so it was effective as a MPSs treatment [98].

Finally, HSp70 protein increases the lysosome stability by modulating the sphingolipid membrane composition. The acid sphingomyelinase deficiency is partially restored by the Hsp70 administration in cells from Niemann-Pick A patients [99].

## 6. Conclusions

The wide range of possible new treatments and therapeutic strategies for LSDs, described in the present article, gives an idea of the great efforts made by the scientific community to provide patients with options against unmet medical needs, which will hopefully be soon available in the clinic.

Nevertheless, there is still margin for improvement in respect to future research in this field. Relevant challenges include the optimization of the available strategies to improve not only lifespan, but also patient’s life quality; and the reduction in the costs associated with these chronic patient’s treatment and follow up. Moreover, therapeutic protocols also need to be improved in future to introduce combinatory therapies and personalized medicine, which will face the specific clinical issues experienced by these patients, who are affected with complex multisystemic and heterogeneous disorders.

## Figures and Tables

**Figure 1 biomolecules-11-01775-f001:**
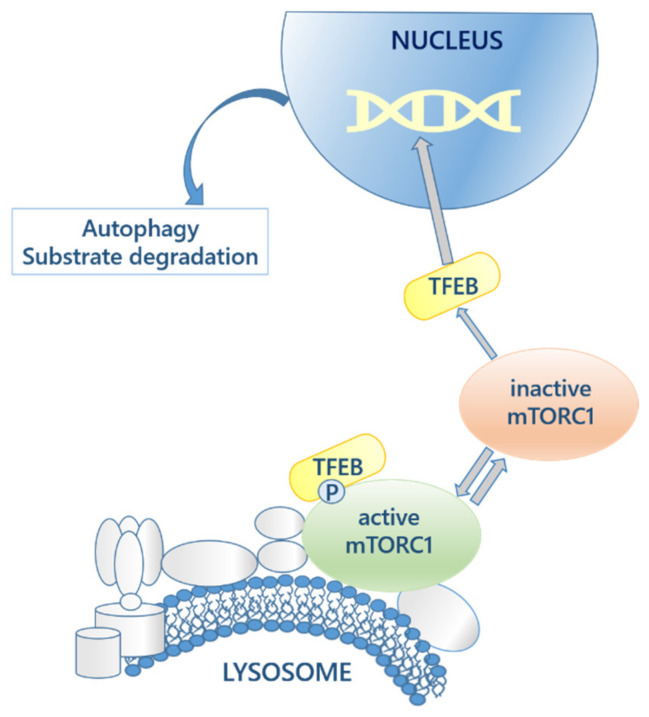
Autophagy activation route and lysosome-nuclei communication. The mTORC1 complex, attached to the lysosomal membrane, controls the activation of TFEB, which is translocated to the nucleus to activate the genes that regulate the autophagic process.

**Figure 2 biomolecules-11-01775-f002:**
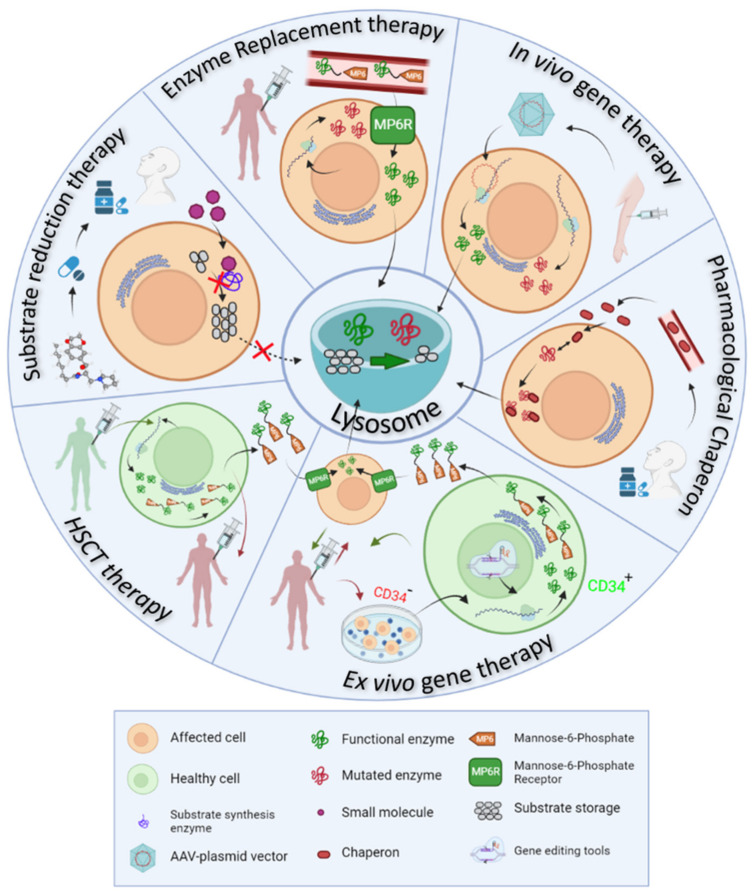
Therapeutic approaches in LSDs. Physiology of the lysosome is at the basis of the therapeutic strategies proposed to treat LSDs. All these approaches aim to restore substrate production/cleavage balance in the lysosome (green semicircle in the center of the image).

**Figure 3 biomolecules-11-01775-f003:**
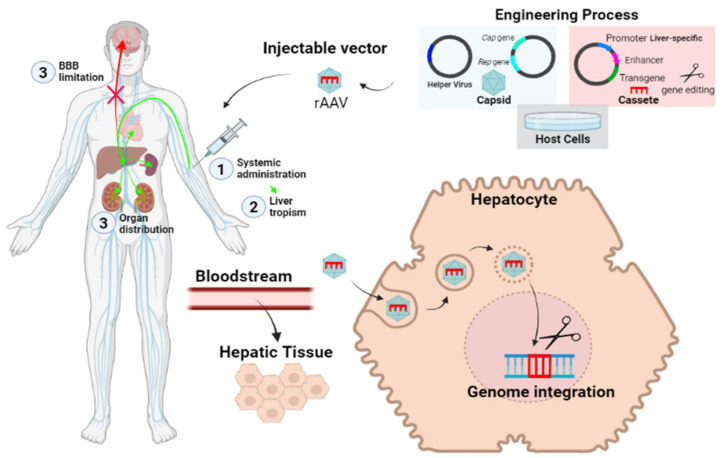
Schematic view of liver targeting in vivo gene therapy.

**Figure 4 biomolecules-11-01775-f004:**
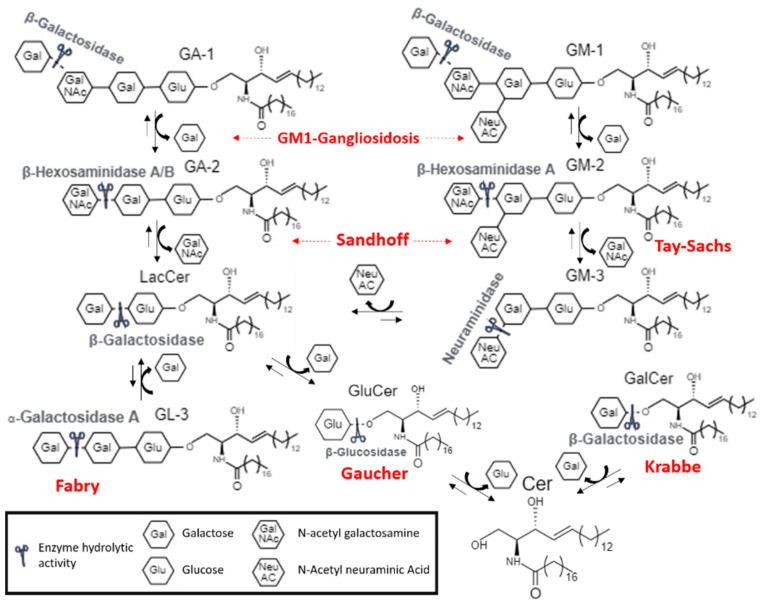
Schematic view of lipid catabolism pathway in the lysosome Enzymes involved in LSDs are indicated.

**Table 1 biomolecules-11-01775-t001:** Available ERT based medicaments for LSDs. In the reference column there are indicated either reference articles or clinical trial numbers referring to www.clinicaltrials.gov (accessed on 15 November 2021) web page.

LSDs	Affected Enzyme	Available ERT	Development Status	Reference
**GD**	*Glucocerebrosidase*	Imglucerase	Approved	[18]
		Velaglucerase	Approved	
		Taliglucerase alpha	Approved	
**FD**	*α-Galactosidase A*	Agalsidase alpa	Approved	[19]
		Agalsidase beta	Approved	NCT03018730
		PRX-102	Phase III	NCT02795676
				NCT03180840
**PD**	*α-Glycosidase*	Alglucosidase alpha	Approved	[20]
		Avalglucosidase alfa	Approved	[21]
		VAL-1221	Phase I-II	NCT02898753
		ATB200	Phase III	NCT03729362
**MPS I**	*α-L-iduronidase*	Laronidase	Approved	[22]
**MPS II**	*Iduronate-2-sulfatase*	Idursulfasa	Approved	[23]
		AGT-182	Phase I	NCT02262338
		JR-141	Approved	[24]
**MPS III** **A** **B**	*Heparan N-sulphatase*	rhHNS	Phase I-II	NCT01299727
	*N-acetyl-glucosaminidase*	BMN250	Phase I-II	NCT02754076
**MPS VI**	*N-acetylgalactosamine-4-sulfatase*	Galsufase	Approved	[24]
**MLD**	*Arilsulphatase A*	HGT1110	Phase I-II	NCT01887938
		TAK611	Phase II	NCT015128 [25]
**NPA and NPB**	*Acid* *Sphingomyelinase*	rh-ASM	Phase I	NCT00410566

**Table 2 biomolecules-11-01775-t002:** Ex vivo gene therapy vectors under development for the treatment of LSDs.

LSD	Drug Name and Trial ID	Action Mechanism	Trial Phase
**GD**	AVR-RD-02 (NCT04145037)	CD34^+^ cells from patients treated with a Lentiviral vector to correct mutations in *GBA*	I/II
**FD**	AVR-RD-01 (NCT03454893)	CD34^+^ cells from patients treated with a Lentiviral vector to correct mutations in *GLA*	I/II
**MLD**	OTL-200 [53] (NCT04283227 III) (NCT03392987) (NCT01560182)	CD34^+^ cells from patients treated with a Lentiviral vector to correct mutations in *ARSA*	I/II
**MPSI**	IDUA LV (NCT03488394)	Lentivirus-based vector to correct defects in *IDUA* gene in CD34^+^ cells	I/II
**MPS II**	L2SN-transduced lymphocytes	Mononuclear cells from blood are extracted from patients and transduced with a retroviral vector which express iduronate-2-sulphatase. Cells are stimulated in order to enrich the lymphocyte T population which are re-implanted in the same patient	I/II

**Table 3 biomolecules-11-01775-t003:** In vivo gene therapy vectors under development for the treatment of LSDs.

LSD	Drug Name and Trial ID	Action Mechanism	Trial Phase
**FD**	FLT190 (NCT04040049)	AAV vector (AAV8) which drive the *GLA* functional gene in the liver	I/II
	4D-310 (NCT04519749)	AAV based vector, which express *GLA* gene under CAG promoter action	I/II
	ST920 (NCT04046224)	vector based on AAV (AAV2/6) which drive the *GLA* functional gene in the liver	I/II
**PD**	SPK-3006 (NCT04093349)	AAV based vector to express *GAA* in liver	I/II
	Raav9-DES-Hgaa (NCT02240407)	Intramuscular AAV9, to express *GAA*	I/II
	AT845 (NCT04174105)	Intravenous AAV8 to express *GAA*	I/II
**MLD**	TYF-ARSA (NCT03725670)	Self-inactivating lentiviral vector injected intracerebrally. The vector transports a correct version of *ARSA* gene.	I/II
**MPSI**	RGX-111 (NCT03580083)	Intracisternal, intracerebroventricular or lumbar puncture of AAV9 which express *IDUA*	I/II
	SB913 (NCT03041324) (NCT04628871)	Permanent expression of iduronidase in hepatocytes, obtained by in vivo genetic editing in albumin locus driven by AAV	I/II
**MPSII**	RGX-121 (NCT0457190)	AAV9-based vector which directs iduronate-2-sulfatase expression	I/II
	SBFIX	Permanent expression of iduronate-2-sulphatase in hepatocytes, obtained by in vivo genetic editing from albumin locus directed by AAV-based vectors	I/II
**MPSIII** **(A, B)**	LYS-SAF302	AAVrh10 intracerebral injection which express *SGSH* gene	I/II
	SAF-301 (NCT02053064)	AAV10 intracerebral injection that express *SGSH* and *SUMF1* genes	I/II
	ABO101 (NCT04655911)	AAV9 which express *NAGLU* gene by intracerebral injection	I/II
**MPS VI**	AAV2/8.TBG.hARSB	AAV8 to express *ARSB* gene in liver	I/II
**Krabbe Disease**	AAVrh10	AAV vector expressing galactosylceramidase is combined with HSCT and delivered intravenously	I/II
	AAV Hu68	Intracisternal AAV vector expressing galactosylceramidase	I/II

**Table 4 biomolecules-11-01775-t004:** Small molecules, approved of under clinical development, for the treatment of LSDs.

LSD	Compound	Strategy	Reference	Trial Phase
**FD**	Migalastat	PC	Approved [73]	
	Lucerastat	SRT	NCT03425539	III
	Venglustat	SRT	NCT02489344	II
**GD**	Venglustat	PC	NCT02843035	II
	Afegostat	PC	NCT00813865 NCT00446550 NCT00433147	II
	Ambroxol	PC	NCT03950050 NCT04388969	II
	Miglustat	SRT	Approved [74]	
	Eliglustat	SRT	Approved [75]	
**PD**	Miglustat	PC	NCT02185651 NCT04327973 NCT04808505	III
	Duvoglustat	PC	NCT01380743 NCT00688597 NCT04327973	II
**Gangliosidosis GM-2**	Venglustat	SRT	NCT04221451	III
	Pyrimethamina	PC	NCT01102686	I/II
**MPS IV**	Odiparcil	SRT	NCT03370653	II
**NPC**	Miglustat	SRT	Approved [76]	
	Arimoclomol	Induces HSP70 chaperone synthesis	NCT02612129	III
	Ostat	Histone deacetylase inhibitor that increase mutant NPC1 protein levels	NCT02124083	I/II

## Data Availability

Not applicable.

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
