# Peer review of "Therapeutic Approaches in Lysosomal Storage Diseases"

_biomolecules, 2021, doi:10.3390/biom11121775_

Round 1
Reviewer 1 Report
The authors aim to review the main therapeutic strategies for LSDs, highlighting possible limitations and future perspectives
This is an useful and important attempt, given the many new treatment options under development for LSD. The authors succeed in a good overview and nice figures and tables.
There would be some points to consider:
- For better readability, it is recommended to improve the language, e.g. by a language editing service.
- For some LSD, hematopoietic stem cell transplantation is an important treatment option, which is not mentioned in this review.
- This review attempts to cover all treatments under development, but several are not mentioned, e.g. for Krabbe disease or MLD
- table 2 and 3 are devided between ex vivo and in vivo approaches. So there is no need to put "in vivo" or "ex vivo" as a column? And table 2 MLD should say "ex vivo"?
- all references for the trials, when mentioned in the text, should be given, e.g. i Dali et al. 2021 for ERT in MLD
- It is stated that there is "no approved gene therapy for LSD", but there is at least for MLD, approved by the EMA in 2020
Author Response
We thank the reviewer for the recommendations and the constructive comments that allowed us to improve the quality of the manuscript.
- English style was extensively revised throughout the whole manuscript. Relevant style changes were highlighted in yellow and new sections and statements were highlighted in light blue.
- Although many new strategies aim to replace allogenic hematopoietic stem cell transplantation, we agree with the referee on the importance of this approach in LSDs with neurological involvement and therefore we added a section dedicated to this therapeutic strategy. (Section 4, page 7, lines 220-236)
- We apologize if we involuntarily omitted some of the available treatments for specific LSDs. The topic of this review is very wide and in constant evolution, so it is easy to forget some important reference. We decide to focus the attention on treatments that are already in the market or in advanced clinical trial phase and to omit mentioning the many gene therapy vectors and small molecules for LSDs in preclinical development, not to exceed the limits of the article. We now found the reference of the recently started trials for Krabbe Disease, which we added to both the text and the tables (Table 3 and page.10 lines 354-359).
ERT and gene therapy treatments for MLD very already cited in the fist version of the manuscript (Table 1, pag.6, line 170 and Table 2 and Pag.8 Lines 274-278).
- We modified the tables, as suggested by the reviewer.
- We added the references of Deli et al. and the other available trial references.
We correct the statement referring to non approved gene therapy for LSD, referring to the recently approved therapy for MLD (Pag.7, line 247). We apologize if we did not realize that this treatment was approved in 2020 and we only mentioned it as a treatment in clinical trial
Reviewer 2 Report
The paper is among many others reviewing LSD treatment. However, LSD review needs to be repeatedly done because the field is moving so fast when it comes to therapeutic developments. No paper written today can encompass all treatments attempted or described even a few months from now. Therefore I believe this paper deserves to be published due to its summary of clinical trials in many modalities of treatment in this heterogenous group of disorders with lots of opportunities to progress. I am concerned about use of English as many sentences need to be re-written with correct English grammar starting with the abstract's last sentence. Also please double check some of the claims that you're making such as "These cells are responsible for the major clinical signs in this condition, nonetheless ERT is not effective in bone involvement or in neurological manifestations of type III GD." In my understanding, bone does benefit from GD ERT treatment.
Author Response
We thank the reviewer for the recommendations and the constructive comments that allowed us to improve the quality of the manuscript.
- English style was extensively revised throughout the whole manuscript. Relevant style changes were highlighted in yellow and new sections and statements were highlighted in light blue.
- We thank the referee for this comment, as we agree with the inappropriate formulation of the sentence written in the first version of the manuscript and relative to the efficacy of ERT in GD. The text as been modified accordingly (Pag.5 Line146-149).
Round 2
Reviewer 1 Report
The revised manuscript has improved considerably.
A paragraph about HSCT has been included. However, the following sentence seems to use incorrect references? And the language ("f") needs to be improved.
"In Krabbe disease and MLD, disease phenotype and stage f progression are of fundamental importance in determining succesull outcomes (46,47)"
In addition, HSCT should be included in figure 2 as well.
Author Response
Reviewer #1’s comments:
The revised manuscript has improved considerably.
A paragraph about HSCT has been included. However, the following sentence seems to use incorrect references? And the language ("f") needs to be improved.
"In Krabbe disease and MLD, disease phenotype and stage f progression are of fundamental importance in determining succesull outcomes (46,47)"
In addition, HSCT should be included in figure 2 as well.
Response: We thank the reviewer again for the constructive suggestions and for appreciating the changes that we introduced in the first revision.
- “The sentence was corrected as follow: “In Krabbe disease and MLD, disease phenotype and stage of disease progression are of fundamental importance in determining succesull outcomes (46,47)”
- The reference list was correctly organized. We apologize if we did not realized that our reference manager did not automatically reorganized the bibliography and maintained the reference list of the original submission, without including the new references.
- HSCT was included in figure 2, as suggested.
Reviewer 2 Report
The authors adequately addressed my concerns
Author Response
We thank the reviewer for appreciating the changes that we introduced to the revised version.